# Knowledge of Osteoporosis and Its Associated Factors among Public Health Professionals in a Municipal Office in Japan

**DOI:** 10.3390/healthcare10040681

**Published:** 2022-04-05

**Authors:** Shino Oba, Naomi Kajiyama

**Affiliations:** 1Graduate School of Health Sciences, Gunma University, 3-39-22 Showa-machi, Maebashi 371-8514, Japan; 2Center for Food Science and Wellness, Gunma University, 3-39-22 Showa-machi, Maebashi 371-8514, Japan; 3Integrated Community Care Support Division, Osaka Nursing Association, 2-2-22 Shiromi, Chuo-ku, Osaka City 540-0001, Japan; n-kajiyama@osaka-kangokyokai.or.jp

**Keywords:** osteoporosis, knowledge, education, public health professional, staff development, nutritionists, nurses, public health

## Abstract

Lifelong efforts to maintain bone health are beneficial for preventing osteoporosis, and public health professionals play an important role in that. The current study aimed to assess the knowledge of osteoporosis among public health professionals in a Japanese municipal office and explored associated factors. A questionnaire was distributed to 124 eligible public health professionals in a municipal office in S City, Japan; in total, 89 individuals who returned it were analyzed. Their knowledge of osteoporosis was assessed using the revised Osteoporosis Knowledge Test, its two subscales, and the Facts on Osteoporosis Quiz, translated into Japanese. To compare the level of knowledge between categories of selected factors, the Wilcoxon rank-sum test or the Kruskal–Wallis test was applied. The mean of correct answers ranged from 70 to 79%, depending on the scale. The knowledge level was significantly higher among registered dietitians than among public health nurses. Higher scores were significantly associated with past learning experience in off-the-job training, with having a family history of osteoporosis, and with having had a past osteoporosis test. Japanese public health professionals were likely to have moderate knowledge of osteoporosis. Several factors were implied to be associated with the knowledge level of osteoporosis.

## 1. Introduction

Broken bones or falls are the third leading known cause of disability among elderly women in Japan, accounting for 12.5% of all causes of disability among the Japanese elderly population in fiscal year 2020 [1]. Osteoporosis increases bone fragility and, consequently, susceptibility to fracture [2]; a public health approach is well suited to the lifelong effort to maintain bone health and prevent osteoporosis [3]. Health professionals are expected to play an important role in the awareness and prevention of the disease at the community level; hence, it is essential for them to have sufficient knowledge of the disease. Nevertheless, only limited studies have paid attention to the knowledge of osteoporosis among public health workers [4], while more studies focused on health professionals in clinical settings [5,6,7,8,9]. Therefore, the aim of the current study was to assess the knowledge of osteoporosis among public health professionals working in a municipal office in Japan. We assessed their knowledge of osteoporosis risk factors, preventive behaviors, and behaviors that promote bone health. We explored their association with factors potentially associated with a higher knowledge level, including past on- and off-the-job training attendance.

## 2. Materials and Methods

### 2.1. Data Collection and Participants

This is a cross-sectional study exploring the knowledge of osteoporosis and its related factors among public health professionals. It was performed from January to February 2019. This study was approved by the Ethics Committee of Gunma University Graduate School of Medicine. The eligible study participants were public health nurses, registered dietitians, and dental hygienists who worked in the municipal office of S City, Japan. A possible association between poor oral health and systemic diseases, including osteoporosis, has been proposed [10], and we included dental hygienists in this study on the basis of their role in promoting oral health among city residents. An explanation of the study’s aim and its procedure was distributed with the self-administered questionnaire to all eligible public health professionals, and only individuals who agreed to participate in the study submitted the questionnaire with their completed consent forms. Of 101 public health nurses, 14 registered dietitians, and 9 dental hygienists, 70, 12, and 7 individuals submitted the questionnaire and the consent form, respectively. Of the participants, three individuals were male. The skewed male–female ratio was assumed to represent the current public health workforce in Japan, and we included both males and females in the analysis.

### 2.2. Measures of Knowledge

The revised Osteoporosis Knowledge Test (OKT) [11] and the revised version of the Facts on Osteoporosis Quiz (FOOQ) [12] were used to assess participants’ knowledge of osteoporosis. The OKT was developed to measure knowledge of risk factors and strategies for osteoporosis prevention. It consists of 32 questions with a score range of 0 to 32. There are also subscales: OKT Nutrition scores range from 0 to 26, OKT Exercise scores range from 0 to 20, and some general information is contained in both subscales. The FOOQ was developed to measure the knowledge of bone health and osteoporosis risk reduction. It consists of 20 questions with a score range of 0 to 20. Both questionnaires in the English language were sent by the authors to the study group, along with authorizations for their use. Both original English questionnaires were translated into the Japanese language by informally adopting a forward- and back-translation process [13], which is described as follows. Two individuals translated the original questionnaire into the Japanese language. One of them is an author of this study, who is native Japanese speaker and studied in the United States, following a 3-year working experience there. The other is a native Japanese speaker who earned a bachelor’s degree in a health science field and had working experience in the pharmaceutical industry. A back translation was conducted by two different professional translators whose native language is Japanese; one of them had a bachelor’s degree in veterinary medicine, and the other has considerable experience as a professional translator in the medical field. Finally, the aforementioned author reviewed the back-translated English questionnaires and compared them to the original questionnaire, analyzed the contents of the forward Japanese translations, and then finalized the Japanese questionnaire.

### 2.3. Study Questionnaire

Questionnaire respondents were asked about age, occupation (public health nurse, registered dietitian, or dental hygienist), position (managerial or not), and number of years working in the municipal office. They were also asked whether they had learning experiences of osteoporosis in college or professional school, at employer-provided training, and/or at training outside of the job. The questionnaire also asked whether they had a working experience related to osteoporosis; whether they had undertaken an osteoporosis test in the past; and whether they had a grandmother, mother, or sister who had osteoporosis. The level of daily physical activity was estimated using the Japanese version of the International Physical Activity Questionnaire and transferred into daily metabolic equivalent minutes per day [14]. Regular intakes of calcium, vitamin K, and alcohol were estimated with validated questionnaires [15,16,17]. The self-reported risk of osteoporosis was measured from the statement, “I have a risk of osteoporosis” with a 5-point Likert scale from “strongly agree” to “strongly disagree.”

### 2.4. Statistical Analysis

To summarize the level of knowledge of osteoporosis, the mean and standard deviations were calculated for the OKT, its subscales, and the FOOQ. The median and mode were also obtained because the test values were not distributed strictly normally. To assess the internal consistency of the knowledge scales, Kuder–Richardson 20 (KR-20) was utilized [18]. KR-20 is a specific case of Cronbach’s coefficient alpha, and they are equivalent when the variables are dichotomous [19,20]. The difference in the knowledge level was assessed by age group, job-related factors, training experience, osteoporosis risk factors, and self-reported osteoporosis risk. For the analysis assessing the difference by profession, dental hygienists were excluded, because the number of participants was insufficient for statistical analysis. The Wilcoxon rank-sum test was applied to compare two categories, and the Kruskal–Wallis test was applied to compare three or more categories. There were a few missing values in the questionnaires, and they were assigned either to the mode or the “do not know/wrong answer” category. All statistical analyses were performed with SAS 9.4 software (SAS, Cary, NC, USA).

## 3. Results

Table 1 shows the participants’ background characteristics and factors with potential association with their knowledge of osteoporosis. Close to 80% of the participants were public health nurses, followed by dietitians (14%) and dental hygienists (8%). About one fifth of participants had female family members with osteoporosis.

Table 2 summarizes the osteoporosis knowledge levels measured by the OKT, its subscales, and the FOOQ. The means of the OKT, the OKT Nutrition, the OKT Exercise, and the FOOQ were 22.3, 14.0, 18.8, and 15.7, respectively; the means of correct answers were 70%, 70%, 72% and 79%, respectively. KR-20 for the OKT total score, nutrition subscale and FOOQ indicate marginally acceptable internal consistency (0.62, 0.59 and 0.55, respectively). Low internal consistency was indicated for the OKT Exercise (KR-20 = 0.45).

Table 3 shows the mean scores of the OKT and the FOOQ and their comparisons according to potentially associated factors as explained in the Methods section. The scores of the OKT and OKT Nutrition were significantly higher among registered dietitians than among public health nurses. Participants who had received past off-the-job training regarding osteoporosis had higher scores on the OKT and OKT Nutrition than did participants without training. Participants who had family members with osteoporosis had significantly higher scores on the OKT and both subscales than did participants without such members. Participants who had taken an osteoporosis test in the past had significantly higher scores on all knowledge scales. No other traits of participants were significantly related to the level of osteoporosis knowledge.

We conducted an ad hoc analysis to explore factors observed to be associated with knowledge of osteoporosis. Half of the registered dietitians reported that they had previously attended off-the-job training regarding osteoporosis, whereas the attendance rate was 8.6% among public health nurses. Of the 14 individuals reporting past attendance at training outside of the job, 13 individuals (92.9%) reported that they had been tested for osteoporosis, whereas 77.3% of individuals who had not attended training had been tested. Among participants who had attended off-the-job training regarding osteoporosis, 35.7% had a family history of osteoporosis, whereas the equivalent prevalence was 17.3% among non-attendees.

## 4. Discussion

The current study found relatively high knowledge of osteoporosis among public health professionals who worked in a municipal office in Japan. Individuals who had attended off-the-job training regarding osteoporosis, who reported a female family history of osteoporosis, and who had taken an osteoporosis test were likely to have higher knowledge of osteoporosis. Registered dietitians were likely to have higher levels of nutritional knowledge of the disease. To our knowledge, this is the first study to assess the knowledge of osteoporosis among public health professionals in Japan, studying both public health nurses and dietitians, using a scale for the nutritional risk factors of osteoporosis.

On average, about 70% or more of the questions were answered correctly on each scale in the current study. The rate of correct answers was about the same as in a study among public health nurses in Taiwan, although that study measured knowledge with a different questionnaire [4]. Several studies measuring knowledge of osteoporosis with the revised OKT among various non-professional individuals have reported low scores; the median was 14.5 among urban Indian adults [21], the mean was 13.5 among diabetes patients in Palestine [22], and the mean was 15.7 among middle-aged female patients in healthcare centers in Poland [23]. Considering that the mean OKT score was 22.3 and the median score was 23 in the current study, it can be assumed that the current study participants had higher levels of osteoporosis knowledge due to their professions. However, we are not able to estimate whether this difference holds true among Japanese lay individuals, since no such data is available. In addition, it should be noted that the majority of the study participants was female, and females are likely to have high knowledge of osteoporosis because they are at high risk of osteoporosis.

The knowledge of osteoporosis among public health professionals in the current study was slightly higher than that reported in a study of professional nurses in Singapore that used the same FOOQ scale. This is partly due to the fact that the current participants included registered dietitians working in a public health setting, as they scored higher on average. They also had high total scores for the OKT and its nutrition subscale, whereas the mean score of the exercise subscale of the dietitians was about the same as that of the public health nurses. Daily diet plays a crucial role in long-term bone health; especially, maintaining an adequate intake of calcium as well as vitamin D, which aids its absorption, may decrease the risk of osteoporotic fractures later in life [24]. Registered dietitians in public health settings can largely contribute to raising awareness of bone health in the community for people of all ages. Designing training programs that emphasize the nutritional aspect of preventing osteoporosis would be necessary for all public health professions. It would be most efficient for registered dietitians and public health nurses to work together; such an inter-professional group would be able to assist in the lifelong effort to improve bone health, osteoporosis prevention, and early treatment of the disease in the community.

In this study, learning about osteoporosis in off-the-job training was associated with more knowledge of osteoporosis, whereas education in schools was not associated with more knowledge, nor was learning at employer-provided training. The results were not coordinated with those from the study of public health nurses in Taiwan [4], in which university education was associated with a higher knowledge level. Participants in the current study may be homogeneous in terms of past educational attainment, as they possessed national certificates of public health nurses, registered dietitians, or dental hygienists issued by the Japanese Ministry of Health, Labour, and Welfare. They also passed the employment selection procedure for the municipal office, which is generally highly competitive in Japan. Research into the prevention of osteoporosis is constantly advancing, and new information is reaching the field. Offering continuing education opportunities to public health professionals would be highly beneficial. Our study cannot identify the specific content of off-the-job training that had been taken by study participants, and studies exploring the details of the training are necessary.

An ad hoc analysis revealed that multiple factors associated with high knowledge of osteoporosis were intercorrelated. Registered dietitians were more likely to have attended off-the-job training regarding osteoporosis than were public health nurses. The number of registered dietitians working in municipal offices was fewer than that of public health nurses, and they may seek opportunities for learning outside of the job setting. Almost all of the individuals who had attended off-the-job training regarding osteoporosis had been tested for osteoporosis in the past. This is probably because bone mineral density was measured during the training session. Furthermore, individuals with a family history of osteoporosis were more likely to have attended the off-the-job training. Health professionals with a family history of osteoporosis may be more willing to learn about the disease. Considering the limited number of participants in each subgroup, we are not able to conduct further analysis with adjustments for these factors. However, the coexistence of these factors may be innate to the participants for the aforementioned reasons, and conducting further analysis while adjusting for them would lead to an overadjustment.

Despite our expectations, the knowledge of osteoporosis and the regular intake of calcium and vitamin K were not observed to be associated among public health professionals. The homogeneity of our participants, who included nationally certified health professionals who were likely to be highly health conscious, may have obscured this particular association. Past intervention studies among lay individuals reported that education regarding osteoporosis improved nutritional intakes [25,26]. Better nutrition intake is one preventive practice guided by the proper knowledge of nutrition. We conducted an additional analysis to examine the association between the prevalence of self-reported diagnosed osteoporosis and knowledge of the disease. However, only five participants reported having osteoporosis, a number too small to obtain a meaningful result.

The foremost limitation of the study is the marginal level of internal consistency of translated knowledge scales. The use of the OKT Exercise may have affected the study result most seriously, with the value of KR-20 less than 0.5, which means an unacceptable level of internal consistency [27]. The length of the OKT Exercise is relatively shorter than the total OKT and the OKT Nutrition, which may have reduced the KR-20 value [28]. Another explanation would be that exercise is known as a protective factor for various health conditions, and thus its health benefit is not limited to osteoporosis. The original English OKT reported high internal consistency for the OKT as well as its subscales, and the translation into Japanese may have introduced heterogeneity. A modification of the OKT Exercise is recommended for further use of this subscale in Japanese subjects, despite the drawback of not being able to make a comparison with other populations. Another limitation was that this study was conducted in one specific municipal office in Japan, and it may lack generalizability. However, all participants were public health professionals who had obtained the national certificates for their professions, and it can be assumed that public health workers in Japan are relatively homogeneous in terms of their knowledge and skills. The study results may be applied to public health workers in Japan, but due to the homogeneity of this group, the ability to generalize the results to the lay population may be limited. Further studies in various regions and populations are recommended to examine the generalizability of the current results. Another limitation of the study is the relatively small sample size, especially in the analysis for occupations, and the findings remain suggestive.

## 5. Conclusions

The current participants—public health workers in a Japanese municipal office—had moderate levels of knowledge of osteoporosis, which was relatively higher than the levels observed in various non-professional populations in previous studies. Knowledge was likely to be higher among registered dietitians than among public health nurses. Learning experience in off-the-job training, undergoing osteoporosis testing in the past, and having a family history of osteoporosis were related to a high level of knowledge about the disease. Studying public health workers in various regions to confirm the current findings is recommended.

## Figures and Tables

**Table 1 healthcare-10-00681-t001:** Background characteristics of public health professionals in a municipal office in Japan.

Variable	No.	%
Age ^1^	37.8 (9.5)	
20s	24	27.0%
30s	28	31.5%
40-older	37	41.6%
Occupations		
Public health nurse	70	78.7%
Registered dietitian	12	13.5%
Dental hygienist	7	7.9%
To be with managerial position ^2^		
Yes	20	22.5%
No	60	67.4%
Length work in the municipal office ^1^	14.1 (10.4)	
-5 years	25	28.1%
>5–15 years	23	25.8%
>15 years or more	41	46.1%
Ever learned about osteoporosis		
In college or professional school	41	46.1%
At the employer-provided training	24	27.0%
In off-the-job training job	14	15.7%
Have job experience related to osteoporosis		
	73	82.0%
Metabolic equivalents (min/day) ^1^	114 (120.7)	
Undertook an osteoporosis test in the past ^2^		
Yes	71	79.8%
No	16	18.0%
Family history of osteoporosis (mother, grandmother, or sister)		
	18	20.2%
Calcium intake four categories		
A or B	13	14.6%
C	33	37.1%
D	32	36.0%
E	11	12.4%
Vitamin K intake, score ^1^	25.2 (11.3)	
Alcohol intake (g) ^1^	9.4 (14.0)	
0	48	53.9%
<20	27	30.3%
≥20	14	15.7%
Self-reported risk of osteoporosis “I have a risk of osteoporosis” ^2^		
Strongly agree	14	15.7%
	21	23.6%
	32	36.0%
	15	16.9%
Strongly disagree	5	5.62

^1^ Values are expressed as mean (standard deviation); ^2^ do not add up to 100% because of missing data.

**Table 2 healthcare-10-00681-t002:** Knowledge of osteoporosis among public health workers.

	Mean	SD	Median	Interquartile Range	Mode
The OKT					
Total (range 0–32)	22.3	3.2	23	5	24
Exercise subscale (0–20)	14.0	2.1	14	3	14
Nutrition subscale (0–26)	18.8	2.7	19	4	19
FOOQ (range 0–20)	15.7	2.1	16	3	17

KR-20 for the OKT, the OKT Exercise, the OKT Nutrition, and FOOQ was 0.62, 0.45, 0.59 and 0.55, respectively. The OKT, The revised Osteoporosis Knowledge Test; FOOQ, Facts on Osteoporosis Quiz.

**Table 3 healthcare-10-00681-t003:** The mean scorers of osteoporosis knowledge scores according to potentially associated factors.

		Osteoporosis Knowledge Test											
	No.	The OKT			The OKT Exercise			The OKT Nutrition			FOOQ		
		Mean	SD	*p* ^3^	Mean	SD	*p* ^3^	Mean	SD	*p* ^3^	Mean	SD	*p* ^3^
All participants		22.3	(3.2)		14.0	(2.1)		18.8	(2.7)		15.7	(2.1)	
Age													
20s	24	21.6	(3.6)		13.5	(2.3)		18.2	(3.2)		15.3	(2.3)	
30s	28	22.5	(3.2)		14.3	(1.9)		18.8	(2.7)		15.8	(2.2)	
40-older	37	22.6	(3.0)	0.52	14.1	(2.1)	0.47	19.3	(2.4)	0.38	15.8	(2.1)	0.74
Occupation ^1^													
Public health nurses	70	21.8	(3.1)		14.0	(2.1)		18.4	(2.6)		15.6	(2.2)	
Registered dietitians	12	24.9	(2.9)	0.002	14.3	(2.2)	0.67	21.0	(2.3)	0.002	16.1	(2.4)	0.41
Managerial Position ^2^													
Yes	20	22.8	(3.0)		14.4	(1.9)		19.0	(2.6)		15.7	(2.5)	
No	60	22.2	(3.2)	0.56	14.0	(2.1)	0.63	18.7	(2.8)	0.85	15.7	(2.1)	0.90
Length work in the municipal office													
0–5 years	25	21.9	(3.8)		13.9	(2.4)		18.4	(3.2)		15.3	(2.3)	
>5–15 years	23	22.5	(2.9)		14.1	(2.0)		18.8	(2.4)		15.8	(2.0)	
>15 years or more	41	22.5	(3.1)	0.82	14.1	(2.1)	1.00	19.1	(2.6)	0.69	15.8	(2.2)	0.79
Ever learned about osteoporosis in college or professional school													
Yes	42	22.0	(3.8)		13.9	(2.4)		18.6	(3.2)		15.7	(2.2)	
No/No response	47	22.6	(2.7)	0.64	14.1	(1.9)	0.92	19.0	(2.3)	0.66	15.6	(2.1)	0.65
Ever learned about osteoporosis at the employer-provided training													
Yes	24	22.6	(2.5)		14.3	(2.1)		19.2	(2.2)		16.2	(2.1)	
No/No response	65	22.2	(3.5)	0.71	14.0	(2.1)	0.56	18.7	(2.9)	0.50	15.4	(2.1)	0.14
Ever learned about osteoporosis from off-the-job training													
Yes	14	24.3	(2.9)		14.5	(2.0)		20.6	(2.4)		15.6	(2.0)	
No/No response	75	21.9	(3.2)	0.02	13.9	(2.2)	0.42	18.5	(2.7)	0.01	15.7	(2.2)	0.75
Job experience related to osteoporosis													
Yes	73	22.3	(3.1)		14.0	(2.0)		18.9	(2.6)		15.8	(2.1)	
No/No response	16	22.3	(3.8)	0.90	14.2	(2.6)	0.71	18.6	(3.1)	0.91	15.2	(2.2)	0.30
Family history of osteoporosis (mother, grandmother, or sister)													
Yes	18	24.2	(2.4)		15.2	(1.9)		20.2	(2.1)		16.2	(2.1)	
No/No response	71	21.8	(3.3)	0.01	13.7	(2.1)	0.01	18.5	(2.8)	0.03	15.5	(2.2)	0.24
Undertook an osteoporosis test in the past ^2^													
Yes	71	22.7	(3.0)		14.2	(2.1)		19.2	(2.4)		15.9	(1.9)	
No	16	20.4	(3.4)	0.01	13.1	(2.2)	0.05	17.1	(3.1)	0.01	14.5	(2.4)	0.03
Physical activity level													
1st tertile	31	22.4	(2.9)		14.0	(1.9)		18.8	(2.5)		15.4	(2.0)	
2nd tertile	29	22.3	(2.8)		14.0	(2.1)		18.9	(2.4)		16.0	(2.4)	
3rd tertile	29	22.3	(4.0)	0.95	14.1	(2.5)	0.81	18.7	(3.3)	0.95	15.7	(2.1)	0.38
Self-reported risk ^2^													
Yes	35	22.2	(2.9)		14.3	(2.0)		18.6	(2.6)		15.7	(2.1)	
No	52	22.2	(3.4)	0.94	13.8	(2.2)	0.47	18.8	(2.8)	0.71	15.5	(2.2)	0.92
Alcohol intake													
0	48	22.7	(3.1)		14.3	(2.1)		19.1	(2.5)		15.7	(2.1)	
<20	27	22.1	(3.4)		13.8	(2.1)		18.6	(3.0)		15.7	(2.1)	
≥20	14	21.4	(3.3)	0.45	13.7	(2.3)	0.58	18.2	(3.0)	0.61	15.5	(2.4)	0.99
Calcium intake four categories													
A or B	13	22.6	(3.7)		14.0	(2.5)		19.3	(2.7)		16.5	(1.5)	
C	33	22.5	(3.0)		14.2	(2.0)		18.9	(2.7)		15.4	(2.4)	
D	32	21.8	(3.3)		13.8	(2.1)		18.4	(2.9)		15.3	(2.0)	
E	11	22.9	(3.2)	0.76	14.5	(2.2)	0.64	19.1	(2.4)	0.84	16.4	(2.2)	0.20
Vitamin K													
1st tertile	30	22.7	(2.9)		14.0	(2.0)		19.3	(2.6)		15.1	(2.0)	
2nd tertile	38	22.2	(3.4)		14.2	(2.1)		18.6	(2.9)		16.0	(2.2)	
3rd tertile	21	22.0	(3.5)	0.77	13.7	(2.4)	0.75	18.6	(2.7)	0.67	15.8	(2.2)	0.11

^1^ Dental hygienists were not analyzed because the number of participants was not sufficient for the statistical analysis; ^2^ individuals with missing values were excluded; ^3^ the Wilcoxon rank-sum test was applied to compare two categories and the Kruskal–Wallis test was applied to compare three or more categories; The OKT, The revised Osteoporosis Knowledge Test; FOOQ, Facts on Osteoporosis Quiz.

## Data Availability

The datasets used and/or analyzed during the current study are available from the corresponding author upon reasonable request.

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
