# Peer review of "Knowledge of Osteoporosis and Its Associated Factors among Public Health Professionals in a Municipal Office in Japan"

_healthcare, 2022, doi:10.3390/healthcare10040681_

Round 1

Reviewer 1 Report

The authors responded appropriately to my comments.  The exercise KR20 on the OKT (Revised 2011,2012) may have been low because exercise is a prevention behavior for conditions other than osteoporosis as well as for osteoporosis. Thus it is not specific for osteoporosis.

Author Response

We agree on this comment, and added a statement in the discussion (line 258-259) as “Another explanation would be that exercise is known as a protective factor for various health conditions, and thus its health benefit is not limited to osteoporosis.”

Reviewer 2 Report

The study is simple and easy to read. The objective is supported by the design. The major limitations were acknowledged. Novelty of the study should be explicitly mentioned. My comments:

  1. The conclusion does not support the results. The authors mentioned that knowledge of osteoporosis was higher than non-professional populations. This comparison was not done in the study, nor defined.
  2. The small sample size obviously is a limitation and while acknowledged it prevented further analysis/adjustments to really identify associated factors. This keeps the findings at best, suggestive.
  3. Subsection 2.2. I'm not sure if its necessary to provide the lengthy career backgrounds of those who translated the questionnaire. Please condense.
  4. Subsection 2.4 can be condensed as well to mention only what was done for missing variables.
  5. The authors may need to expound if sexual differences in knowledge about osteoporosis is anticipated, since the participants are skewed in favor of females. This may have contributed to the high level of knowledge since they are the population at higher risk.

Minor

  1. Please use dietitian throughout the paper instead of dietician, as this is the preferred term internationally.
  2. Please be consistent with the use of Palatino linotype. Tables and references were presented in Arial narrow.
  3. Title for subsection 2.3 can be changed to ''Study Questionnaire"

Author Response

1. The study is simple and easy to read. The objective is supported by the design. The major limitations were acknowledged. Novelty of the study should be explicitly mentioned. My comments:

The conclusion does not support the results. The authors mentioned that knowledge of osteoporosis was higher than non-professional populations. This comparison was not done in the study, nor defined.

Reply: We agree on this comment. Knowledge current participants was not directly compared to other non-professional populations. The difference was interpreted from the previous studies. Having this comment, we clarified this point in conclusions section, line 277.  

2. The small sample size obviously is a limitation and while acknowledged it prevented further analysis/adjustments to really identify associated factors. This keeps the findings at best, suggestive.

Reply: We agree on this comment. It was inevitable in the current study design yet it was unfortunate. We added at the end of the limitation section (Line 272-3) that the findings remain suggestive (rather than definitive).

3. Subsection 2.2. I'm not sure if its necessary to provide the lengthy career backgrounds of those who translated the questionnaire. Please condense.

Reply: We agree on this suggestion. We condensed the descriptions in line 77-84.

4. Subsection 2.4 can be condensed as well to mention only what was done for missing variables.

Reply: Having this suggestion, we simplified the description in Subsection 2.4

5. The authors may need to expound if sexual differences in knowledge about osteoporosis is anticipated, since the participants are skewed in favor of females. This may have contributed to the high level of knowledge since they are the population at higher risk.

Reply: Unfortunately, we are not able to further explore the male-female difference as the number of male participants was extremely small, although we agree on this comment. We added the statement that the relatively high knowledge in the current participants could be due to a female dominant distribution in Line 196-198.

Minor

6. Please use dietitian throughout the paper instead of dietician, as this is the preferred term internationally.

Reply: Thank you for the important advice. We revised the manuscript and corrected all of them.

7. Please be consistent with the use of Palatino linotype. Tables and references were presented in Arial narrow.

Reply: We leave this issue to the editorial office because it seems like the font was converted by the office.

8. Title for subsection 2.3 can be changed to ''Study Questionnaire"

Reply: We changed it as suggested. Thank you for another important advice.

Reviewer 3 Report

The abstract is concise and well written. The research methodology on the problem definition is appropriate and applied properly. The paper is easy to read and free from grammatical or spelling errors, but some suggestions are recommended.

1-In the “Introduction” section, it is suggested to cite, to increase and to discuss more literature relevant to topics of the incidences of osteoporosis and bone mass density. Some references are recommended:

-Mathematical Model for the Assessment of Fracture Risk Associated with Osteoporosis. Numerical Analysis and Applied Mathematics (ICNAAM 2012), Vols A and B, Book Series: AIP Conference Proceedings, 1479: 814-817, 2012. DOI:10.1063/1.4756262

- Bone fragility in postmenopausal women: a preliminary study. International Journal of Medical Engineering and Informatics, 4(4): 387-397, 2012. DOI: 10.1504/IJMEI.2012.050279

2- In the topic ‘’Data collection’’ authors refer how the study information was done. But in my opinion a flow chart of participant´s recruitment (or investigation and treatment…by groups), could be appear in this topic.

3- In my opinion, authors need to introduce, that the study followed ethical guidelines.

4- A topic about ‘’Strengths and Limitations’’ due several strengths and weaknesses result from the retrospective nature of this study could be included.

5- In my opinion, the definition of some variables / abbreviations could appear at the end of the table 3.

6-Please increase the conclusion section, with the limitations of your study (all that you may find).

Author Response

The abstract is concise and well written. The research methodology on the problem definition is appropriate and applied properly. The paper is easy to read and free from grammatical or spelling errors, but some suggestions are recommended.

1-In the “Introduction” section, it is suggested to cite, to increase and to discuss more literature relevant to topics of the incidences of osteoporosis and bone mass density. Some references are recommended:

-Mathematical Model for the Assessment of Fracture Risk Associated with Osteoporosis. Numerical Analysis and Applied Mathematics (ICNAAM 2012), Vols A and B, Book Series: AIP Conference Proceedings, 1479: 814-817, 2012. DOI:10.1063/1.4756262

- Bone fragility in postmenopausal women: a preliminary study. International Journal of Medical Engineering and Informatics, 4(4): 387-397, 2012. DOI: 10.1504/IJMEI.2012.050279

Reply:  We totally agree that it is very important to introduce the background of the study. However, we are willing to focus on the role of public health professionals for the disease prevention. Introducing wide range of risk factors of osteoporosis would widen the scope of this paper too much. We leave the decision on this issue to the editorial board members. I personally found that these two suggested papers are very interesting to read, which motivates me to plan the next study.  

2- In the topic ‘’Data collection’’ authors refer how the study information was done. But in my opinion a flow chart of participant´s recruitment (or investigation and treatment…by groups), could be appear in this topic.

Reply: Thank you for the comment. The flow chart would be helpful, but this is a simple cross-sectional study and it may not be needed.

3- In my opinion, authors need to introduce, that the study followed ethical guidelines.

Reply: We submitted the study protocol to the University Ethics Committee before conducting this study and obtained the approval. Informed consent was obtained from all the participants. The information is provided in the first paragraph of the Materials and Methods section.

4- A topic about ‘’Strengths and Limitations’’ due several strengths and weaknesses result from the retrospective nature of this study could be included.

Reply: The limitation of the study and our rebuttal are stated in the second last paragraph.

5- In my opinion, the definition of some variables / abbreviations could appear at the end of the table 3.

Reply: We spelled out the OKT and FOOQ in the foot note in Table 2 and 3. Thank you for the helpful suggestion.

6-Please increase the conclusion section, with the limitations of your study (all that you may find).

Reply: The limitations of this study are listed in the paragraph right before the conclusion section.

This manuscript is a resubmission of an earlier submission. The following is a list of the peer review reports and author responses from that submission.

Round 1

Reviewer 1 Report

No novelty research in the field. The obtained results were expected because the study was conducted on a population with an obvious knowledge of the investigated condition. The used tests (OKT and FOOQ) were used after translation, but it is well known that a new implementation of every translated forms should be first done for the language used, as translation- induced biases are a frequent issue. You should first validate and test/implement the translated  version in Japanese and then use it in a study. 

Poor English, with unclear, confusing phrasing and multiple repetitions. 

Reviewer 2 Report

The study titled "Knowledge of Osteoporosis and Its Associated Factors Among 2 Public Health Professionals in a Municipal Office in Japan" explores the awareness and knowledge of osteoporosis among public health professionals in a Japanese municipal office using OKT and FOOQ. The assessment of knowledge about osteoporosis and related problems has an important implication in its tracking, management and alleviation in general public.

Comments.

  1. Experimental methodology is well explained.
  2. Homogeneity of the cohort may have skewed the outcomes of the study.
  3. An results of knowledge of osteoporosis among public health professionals may be discussed in light of similar studies among healthcare professionals and general public. This may help in better design of training programs.
  4. An additional analysis or brief discussion about correlation between osteoporosis awareness among public health professionals and incidence of osteoporosis (diagnosis and mitigation) may be helpful in informing the utility of training programs and their impact on improving the public health quality.

Reviewer 3 Report

Osteoporosis is a concern as our population ages and Asian women, such as the Japanese population in this study, are at relatively high risk.  Thus this study is important as it addresses the osteoporosis knowledge and related factors of health professionals as they have the opportunity to interact and educate those at risk.  Some of the references are old that deal with osteoporosis. 

The OKT (revised 2011, 2012) needs to be stated as such and the citation should be either:

Developed by Katherine Kim PhD, Mary Horan PhD, and Phyllis Gendler PhD (1991). Grand Valley State University, with support from the Grand Valley State University Research Grant-in-Aid. Revised by Phyllis Gendler PhD, Cynthia Coviak PhD, Jean Martin PhD, and Katherine Kim PhD (2011, 2012). Question 26 was developed as an addition to the Osteoporosis Knowledge Test by Pamela von Hurst (2006). 

or the publication of its development:

Gendler, P.E., Coviak, C.P., Martin, J.T., Kim, K.K., Dankers, J.K., Barclay, J.M., Sanchez, T.A. (2014). Revision of the Osteoporosis Knowledge Test: Reliability and Validity. Western Journal of Nursing Research, online June 11, 2014, doi: 10.1177/0193945914537565

The explanation of the instrument should state that it includes some general information which is contained in both subscales.  That would explain why the registered dieticians scored significantly higher on both the whole scale and the Nutrition subscale.  There was reference to Appendix 2 which was not included in the documents I received so I could not evaluate it.

The Statistical analysis appeared appropriate except for the testing of reliability for the OKT (revised 2011, 2012).  Since the responses to the items are either correct or not correct (dichotomous) a KR20 would be an appropriate analysis.  Depending on the statistical package used, that may have been what was correctly done because of the data. Of relevance is that the registered dieticians scored significantly higher on the nutrition subscale which speaks to validity.